# Nucleosome positioning stability is a modulator of germline mutation rate variation across the human genome

Cai Li [1,4✉] & Nicholas M. Luscombe [1,2,3]

Nucleosome organization has been suggested to affect local mutation rates in the genome. However, the lack of de novo mutation and high-resolution nucleosome data has limited the investigation of this hypothesis. Additionally, analyses using indirect mutation rate measurements have yielded contradictory and potentially confounding results. Here, we combine data on >300,000 human de novo mutations with high-resolution nucleosome maps and find substantially elevated mutation rates around translationally stable ('strong') nucleosomes. We show that the mutational mechanisms affected by strong nucleosomes are low-fidelity replication, insufficient mismatch repair and increased double-strand breaks. Strong nucleosomes preferentially locate within young SINE/LINE transposons, suggesting that when subject to increased mutation rates, transposons are then more rapidly inactivated. Depletion of strong nucleosomes in older transposons suggests frequent positioning changes during evolution. The findings have important implications for human genetics and genome evolution.

[1] The Francis Crick Institute, London NW1 1AT, UK. [2] Okinawa Institute of Science & Technology Graduate University, Okinawa 904-0495, Japan. [3] UCL Genetics Institute, University College London, London WC1E 6BT, UK. [4] Present address: School of Life Sciences, Sun Yat-sen University, Guangzhou 510275, China. ✉email: licaigd@gmail.com

Germline de novo mutations, which can be passed to offspring, are the primary source of genetic variations in multicellular organisms, contributing substantially to biological diversity and evolution. De novo mutations are also thought to play significant roles in early-onset genetic disorders such as intellectual disability, autism spectrum disorder, and developmental diseases[1,2]. Thus, investigating the patterns and genesis of de novo mutations in the germline is important for understanding genome evolution and human diseases.

Germline and somatic mutation rates vary across the human genome at diverse scales ranging from nucleotide to chromosomal resolution[3,4]. Studies revealed factors linked to local mutation rate variation, including sequence context[5], replication timing[6], recombination rate[7–9], DNA accessibility[10], and histone modifications[5,11]. However, genomic features identified so far explain less than 40% of the observed germline mutation rate variation (at 100 Kb–1 Mb resolution)[12,13]. Therefore, important factors remain to be found. Moreover, due to the limited availability of de novo mutation datasets, studies focused on coarse-grained mutation rate variation (typically ≥ 1 Kb windows for germline data), or used within-species polymorphisms and interspecies divergence whose observations are potentially confounded by natural selection and other evolutionary processes.

Moreover, the underlying mutational processes causing the observed mutation rate variation are poorly understood, though recent studies have highlighted the contributions of error-prone replicative processes[14–18] and differential DNA repair efficiencies[10,19–21]. Despite these advances, many details of the molecular mechanisms associated with mutation rate variation remain to be uncovered, particularly in the germline.

Here, we focus on the role of nucleosomes in modulating germline mutation rates. Chromatin is considered important because structural constraints could affect the mutability of genomic sequences[22]. Nucleosome organization (including positioning and occupancy) has been reported as a significant factor in humans and other eukaryotes[5,17,23–26]. Studies in different lineages[17,23,24] reported increased substitution rates around the centers of nucleosomal sequences and increased insertion/deletion rates in linker DNA. However, there are also disagreements between published studies. For example, Michaelson et al.[5] suggested that high nucleosome occupancy tends to suppress de novo mutations, but Smith et al.[13] found that a comparative analysis using datasets from different studies resulted in opposing conclusions. Due to few available de novo mutations for humans, analysis of many studies was based on variant data from within-species polymorphisms or interspecies divergence, which can be affected by natural selection and nonadaptive processes such as GC-biased gene conversion. Furthermore, because of the limitation of available nucleosome maps, some previous studies treated all annotated nucleosomes equally, ignoring the diverse contexts in which they form. Therefore, combined with the scarcity of de novo mutation datasets, the effects of nucleosome organization on germline mutation rate variation, particularly at high resolution, remain to be elucidated.

Here, we take advantage of the rapid increase in the number of de novo mutation datasets and better understanding of nucleosome organization in the human genome to perform a systematic analysis of this topic. We reveal increased mutation rates around strongly positioned nucleosomes and suggest that low-fidelity replication, insufficient mismatch repair (MMR), and increased double-strand breaks (DSB) are potential mutational mechanisms linked to strong nucleosomes. Finally, we show that strong nucleosomes are particularly enriched in young transposons, implying an interesting relationship between nucleosomes, transposons, and mutation rates.

## Results

**Datasets used for analysis.** We used >300,000 human de novo single-nucleotide variants (SNVs) and >30,000 short insertions/deletions (INDELs), having removed genomic regions that could confound downstream analysis (Fig. 1a, Supplementary Fig. 1a; see Methods). Most data come from three large-scale trio sequencing projects that contribute about 100,000 mutations each[27–29]. We also examined extremely rare variants (allele frequency ≤ 0.0001) from the gnomAD database[30], which are approximated to de novo mutations because they are thought to undergo limited selection and nonadaptive evolutionary processes[31].

Nucleosome positioning on the genome is described by the translational setting, which defines the location of the nucleosomal midpoint (also called "dyad") and the rotational setting, which defines the orientation of the DNA helix on the histone surface[32]. Using MNase-seq measurements, Gaffney et al.[32] identified ~1 million strong nucleosomes that adopt highly stable translational positioning across seven lymphoblastoid cell lines. Rotationally stable nucleosomes were previously identified from DNase-seq measurements across 43 cell types[33], covering 892 Mb of the genome. There is a ~50 Mb overlap between regions bound by strong nucleosomes and rotationally stable nucleosomes. Using these data, we classified the genome into three groups of regions (Fig. 1b; sex chromosomes excluded): (i) those containing translationally stable, "strong" nucleosomes (198 Mb); (ii) those with rotationally but not translationally stable nucleosomes (796 Mb); and (iii) all other non-N base genomic regions (1703 Mb). West et al.[34] reported that with the exception of a few specific loci such as transcription start sites, overall nucleosome positioning varies little between cell types. None of the nucleosomal datasets were produced using germ cells, therefore as a precaution we excluded nucleosomes that differ in positioning between cell types (~23 Mb; see Methods).

**De novo SNVs and INDELs are enriched in strong nucleosomes.** Genomic regions containing strong nucleosomes have ~30% more de novo SNVs (Fig. 1c) and ~15% more de novo INDELs (Fig. 1d) than expected (without considering the sequence composition and other genomic features). Similar increases are also apparent for extremely rare variants (Supplementary Fig. 1b, c), though effect sizes are smaller than for de novo mutations, probably due to the fact that highly mutable sites are underrepresented among extremely rare variants[35]. When dividing strong nucleosomes by translational stability (based on S(i) scores from Gaffney et al.), we found that those with higher translational stability scores also exhibit higher mutation rates (Fig. 1c, d). These results suggest that translational stability, a previously unappreciated aspect, is associated with local variation in mutation rates, which may affect not only strong nucleosomes but also other parts of the genome. Regions containing rotationally stable nucleosomes, in contrast, are slightly depleted of both mutation types; we did not perform further analysis on this, as effects of rotational positioning have been comprehensively discussed recently by Pich et al.[26]. A more detailed view with meta-profiles clearly depicts increased SNV and reduced INDEL densities around dyad regions of strong nucleosomes compared with flanking linker regions (Fig. 1e), in line with observations made using polymorphism data[24].

Interestingly, ~80% of strong nucleosomes overlap with repeats (Fig. 1f, Supplementary Fig. 1d), especially SINE/Alu (~44%) and LINE/L1 elements (~26%). Genetic variations in repeats are traditionally hard to detect because of poor mappability and so analyses have tended to be cautious in calling variants, resulting in many false negatives (though, few false positives; see Lee and

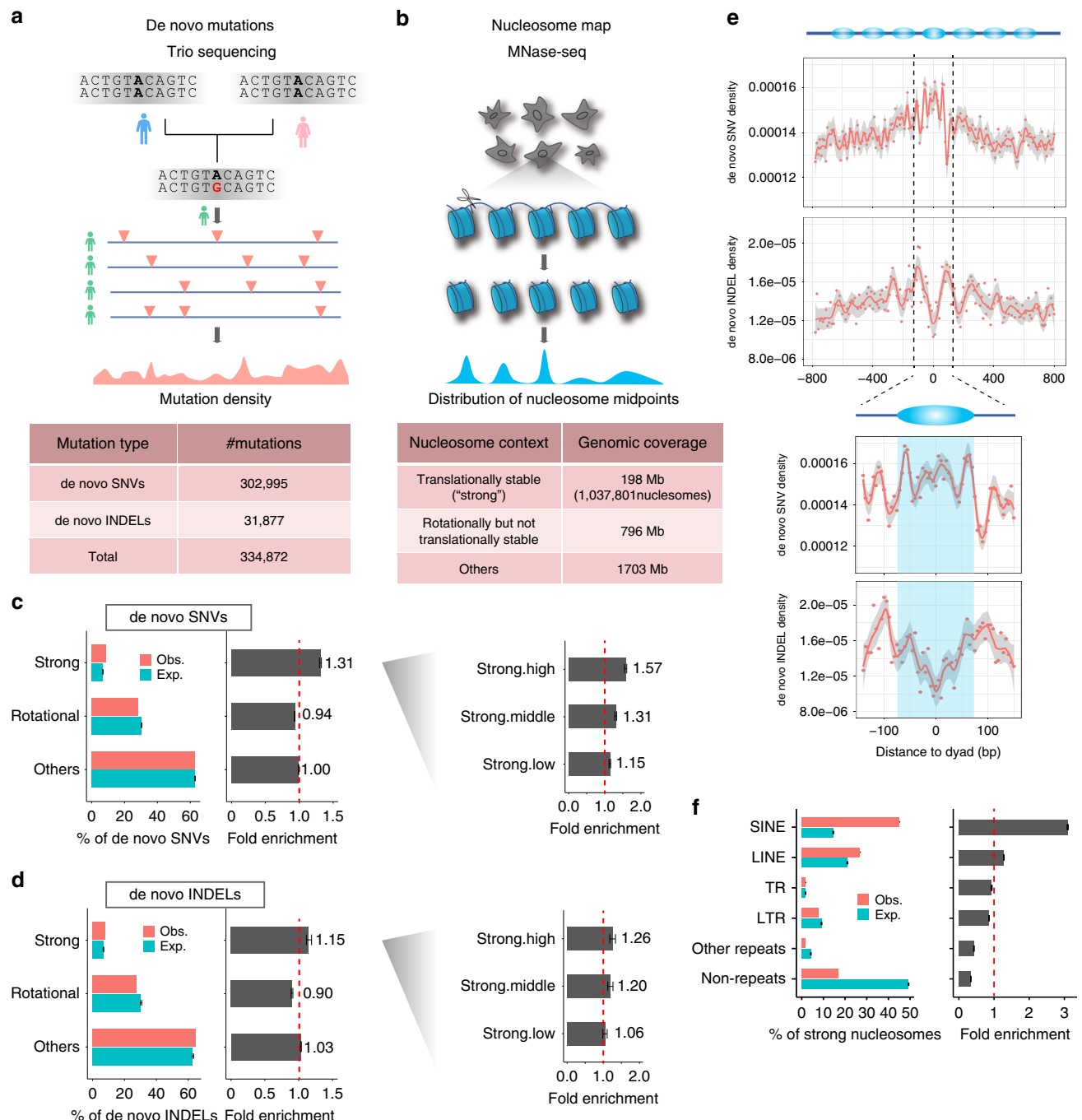

**Fig. 1 De novo mutations are enriched in strong nucleosomes. a** Summary of germline de novo mutation data included in study. **b** Summary of nucleosome positioning data analyzed in study. Observed vs. expected occurrence and fold enrichments of de novo **c** SNVs and **d** INDELs in the three different nucleosome contexts. Right-hand panel subdivides strong nucleosomes according to high, medium, and low translational stabilities. Error bars depict 95% confidence intervals. **e** Top panels, meta-profiles of de novo SNV and INDEL densities relative to position of strong-nucleosome dyads (negative values in x axis representing upstream positions). Bottom panel, same meta-profiles zoomed into the middle nucleosome. **f** Fold enrichment of strong nucleosomes in different repeat elements: short interspersed nuclear element (SINE), long interspersed nuclear element (LINE), tandem repeat (TR), and long terminal repeat (LTR). Source data are provided as a Source Data file.

Schatz[36]). Therefore, the above observations may underestimate the true enrichment of de novo mutations in strong nucleosomes. We subdivided strong nucleosomes into three groups: (i) Alu-associated, (ii) L1-associated, and (iii) others. Alu-associated nucleosomes display increased SNV rates around the dyads, as seen in the meta-profiles for all strong nucleosomes (Supplementary Fig. 1e), whereas non-Alu nucleosomes show increased SNV rates ~60 bp away from the dyads, close to the nucleosome

edges. Such differences may be due to the different local sequence composition (discussed in next section). In contrast, the patterns of INDEL densities are relatively similar among different groups (Supplementary Fig. 1e).

**Genome-wide assessment and controlling for other factors.** Many factors are associated with mutation rate variation. One of

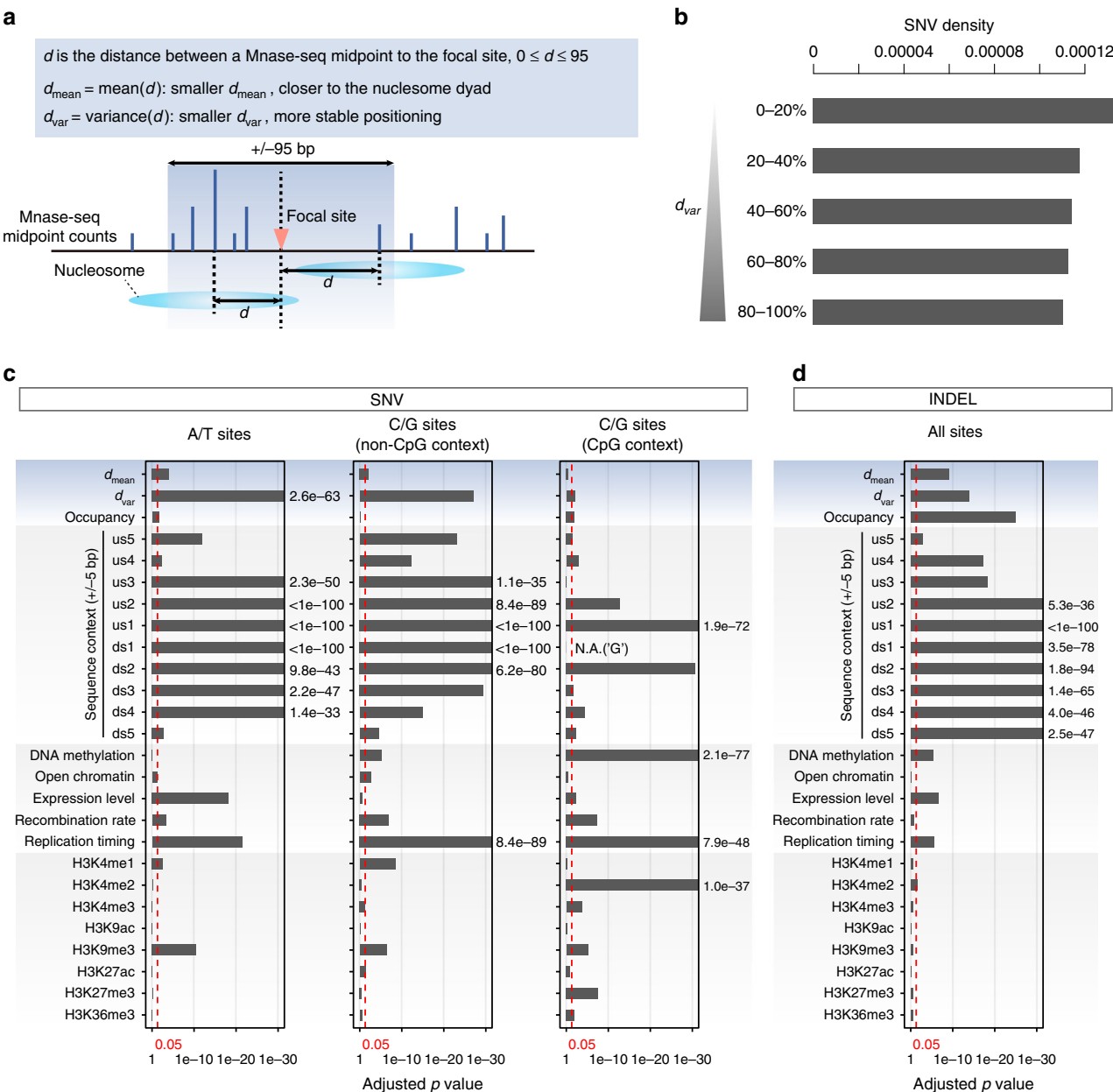

**Fig. 2 Evaluating the contribution of nucleosome organization to mutation rate variation. a** Schematic diagram describing two nucleosome positioning-related variables ($d_{mean}$ and $d_{var}$) relative to a given genomic position. **b** Classifying the genome into five equal portions by $d_{var}$ and calculating the SNV densities. **c, d** Independent statistical significance of potential contributing factors to mutation rate variation, having controlled for other factors; **b** for SNVs and **c** INDELs. Tests for SNVs were performed separately at A/T and C/G sites (non-CpG and CpG contexts, respectively). Vertical red lines indicate the threshold for statistical significance (0.05). The $p$ values were from the likelihood-ratio tests and were adjusted for multiple testing with Benjamini–Hochberg correction. us upstream; ds downstream. Source data are provided as a Source Data file.

the most important is local sequence context — for example, CpG sites are known to be highly mutable and CpG density profiles correlate well with mutation rate profiles in strong nucleosomes (Supplementary Fig. 1e). Functional factors like DNA methylation, histone modification, chromatin accessibility, replication timing, and recombination rate are also relevant. Therefore, to systematically assess the contribution of nucleosomes to mutation rate variation across the whole genome, we defined variables for measuring nucleosomes and used a logistic regression framework to control for potential confounding factors (Fig. 2).

We defined three variables to quantify nucleosomal properties relative to a specific nucleotide position in the whole genome. Two relate to translational positioning: $d_{mean}$, the mean distance

between the focal position and the midpoints of mapped MNase-seq fragments (maximum distance of 95 bp), and $d_{var}$, the variance of these distances (Fig. 2a). A smaller $d_{mean}$ means that a nucleotide position is closer to nucleosome dyads and a smaller $d_{var}$ indicates that the nucleosomes around it are more translationally stable. Our modeling used $d_{var}$ instead of the score $S(i)$ defined in Gaffney et al. to measure positioning stability, because $S(i)$ was designed for dyad positions rather than any position in the genome. When dividing the genome into five equal portions by $d_{var}$, we observed a negative relationship between $d_{var}$ and SNV density, suggesting that $d_{var}$ behaves like $S(i)$ and affects the mutation rates genome-wide (Fig. 2b). As the relationship between $d_{mean}$ and SNV rate is nonlinear, we defined

$d_{mean}$ a categorical variable binned into five intervals (Fig. 1e, Supplementary Fig. 1e; see Methods). The third variable is nucleosome occupancy calculated as a normalized per-base MNase-seq fragment coverage (see Methods). Other factors considered are local nucleotide sequences (±5 bp of the focal site, "−" for upstream) and functional genomic measurements in human germ cells or other cell types if no available germ-cell data (see Methods). $d_{var}$ has a relatively weak but statistically significant correlation with many of these factors, suggesting nonindependence (Supplementary Fig. 2).

To assess the contribution of each factor to local mutation rates, we compared a full logistic regression model encompassing all variables against reduced models missing individual variables using likelihood-ratio tests; the reported $p$ values indicate how significant a factor is associated with mutation rate variation, having controlled for other factors (Fig. 2c, d; see Methods). For SNVs, we tested A/T (comprising A > C, A > G, and A > T mutations), CpG, and non-CpG C/G sites separately (comprising C > A, C > G, and C > T mutations for C/G sites; Fig. 2c), whereas they were pooled for INDELs.

Our statistical framework recapitulates reported observations (Fig. 2c, d, Supplementary Fig. 3). In agreement with previous studies[31], local sequence context is the biggest contributor to local mutation rate variation (Fig. 2c, d), with effect sizes generally declining with increasing distance from the surveyed site. DNA methylation and H3K9me3 are two common epigenetic marks associated with mutation rate variation in general[11], whereas H3K4me1, H3K4me2, H3K4me3, H3K27me3, and H3K36me3 are linked with specific mutation types. Replication timing has highly statistically significant associations with both SNVs and INDEL mutation types. Recombination rate, open chromatin (measured by ATAC-seq), and gene expression level are also associated with specific mutation types.

Turning to nucleosomal properties, translational stability ($d_{var}$) is associated with elevated mutation rates at A/T, non-CpG C/G, and CpG sites, with the first two showing the greatest significance. INDELs also show similar effects, though the higher $p$ values compared with SNVs could partly be due to the smaller sample size. Examining specific SNV mutation types, $d_{var}$ is significantly associated with all A/T and C/G mutations (Supplementary Fig. 3), except for CpG > TpG (adjusted $p = 0.09$). The regression coefficients for $d_{var}$ are always negative (i.e., nucleosome variability is anticorrelated with mutation rate, see coefficients in Supplementary Data 1), indicating that translational stability is positively associated with mutation rates thus corroborating the patterns observed in Fig. 1. We also calculated the McFadden's pseudo $R^2$, which measures the explained variance by $d_{var}$ in the models, but note that currently there is no widely accepted measurement of explained variance for logistic regression. We found that the differences in pseudo $R^2$ between full and reduced models without $d_{var}$ range from 0.07 to 8.88% of the full-model pseudo $R^2$ (median = 1.56%, Supplementary Data 2), suggesting unignorable effects of $d_{var}$.

As expected from Fig. 1, the mean distance to dyads, $d_{mean}$, also displays statistically significant associations with mutations rates at A/T and C/G sites (Fig. 2c, d). Finally, nucleosome occupancy is also statistically significant; in contrast to the positioning variables, here the effect is much larger for INDELs than SNVs (Fig. 2c, d; INDELs, adjusted $p = 9.8e-26$; SNVs, adjusted $p = 0.74, 0.02,$ and $0.01$). The regression coefficients of occupancy are negative for SNVs at A/T sites, but positive for SNVs at CpG sites (Supplementary Data 1), suggesting that occupancy can have opposing effects on mutability depending on sequence context.

Nucleosome positioning stability is at least partly determined by the occupied DNA sequence and thus its effects on mutation rates to some degree can be attributed to the associated sequence (this also applies to other reported factors such as replication timing). We acknowledge the limitation that logistic regression model cannot assess all higher-order interactions among the long stretches of nucleotides that guide nucleosome positioning. It is also impossible to evaluate all possible interactions between local sequences and many functional features. Nonetheless, we achieved similar statistical significance for translational stability after including nonadditive two-way interaction effects for ±5 nucleotides and the 7-mer mutability estimates from Carlson et al. in regression models (see Methods; Supplementary Fig. 4a, b).

Since many strong nucleosomes are associated with repeat elements, we added repeat status as a predictor in the regression model (see Methods). We still observed strong statistical significance for translational stability after considering repeat status (Supplementary Fig. 4c), suggesting that translational stability is independently associated with mutation rate variation. We also tested repeat and nonrepeat regions separately, and translational stability is statistically significant in most tests (Supplementary Fig. 4d).

Taken together, the logistic regression modeling analysis recapitulated known factors and confirmed the independent contribution of nucleosome translational stability to mutation rate variation across the genome.

**Mutational signature analysis.** Having established an association between mutation rate and nucleosome translational stability, we next sought to identify mutational mechanisms that might explain it. As an initial screen, we compared the Catalogue of Somatic Mutations in Cancer (COSMIC) mutational signatures[37] for de novo mutations within strong nucleosomes and those in genomic background. Mutational signatures were originally developed to infer the mutational processes underlying cancer progression by combining the relative frequencies of 96 possible mutation types (six types of single-nucleotide substitutions C > A, C > G, C > T, T > A, T > C, and T > G, each considered in the context of the bases immediately 5′ and 3′ to each mutated base).

We first consider the relative frequencies of the 96 mutation types in the whole genome and strong nucleosomes in different repeat contexts (Fig. 3a). The results account for background differences in tri-nucleotide frequencies between these regions (see Methods). Several mutation types display distinct frequencies in strong nucleosomes, suggesting differences in the underlying mutational processes. For instance, six out of 16 T > C mutation types are more prevalent in strong nucleosomes and different repeat-based subgroups display distinct C > T mutation frequencies. L1-associated strong nucleosomes tend to show the most similar mutation frequencies to genomic background, whereas the "Others" group shows the most changes, perhaps reflecting the heterogeneity of constituent genomic regions.

Next, we applied the MutationalPatterns software[38] to calculate the contribution of COSMIC mutational signatures to different sets of de novo SNVs. Three major signatures (Signatures 1, 5, and 16) are present in all tested groups (contributing 87.7% for the whole-genome group, 77.0–84.5% for strong-nucleosome groups; Fig. 3b). Four signatures (Signatures 5, 12, 20, and 26) show increased contribution (>1%) to the "all strong-nucleosome" group relative to the genomic background. The aetiologies of Signatures 5 (~7% increase in strong-nucleosome regions) and 12 (2.2% increase) are currently unknown according to the COSMIC website, but a recent study[39] suggested that Signature 5 is likely associated with POL θ-mediated mutagenesis and DSB repair. Signatures 20 (1.3% increase) and 26 (1.2% increase) are associated with DNA MMR. There are further differences in associated signatures among strong-nucleosome-associated SNVs in different repeat contexts ("Alu", "L1", and "Others"; Fig. 3b),

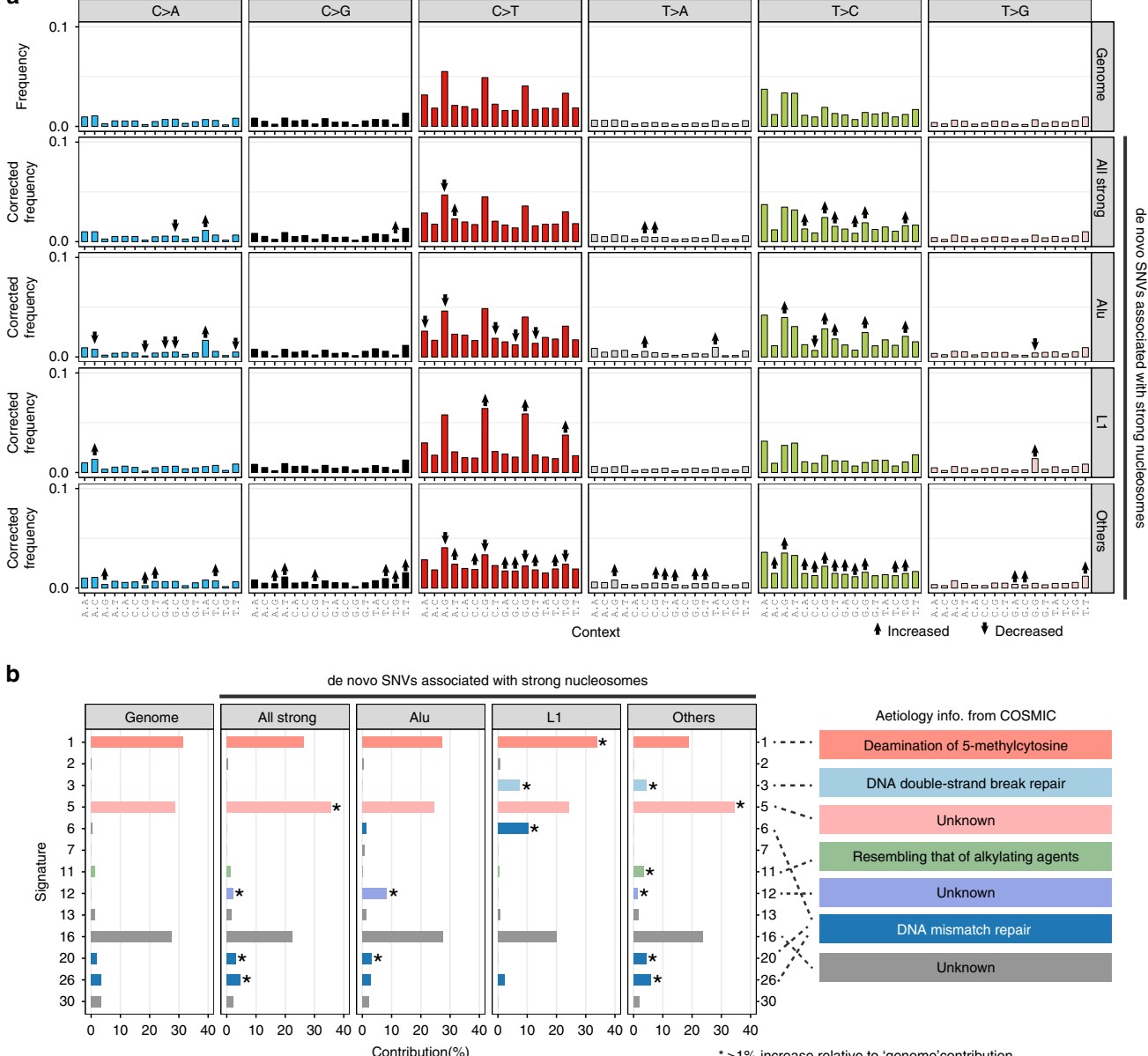

**Fig. 3 De novo SNVs in strong nucleosomes display distinct mutation type frequencies and COSMIC mutational signatures. a** Frequencies of 96 mutation types among de novo SNVs; six nucleotide substitutions in the context of the bases immediately 5′ and 3′ of the mutated site. SNVs are grouped into those overlapping strong nucleosomes and those elsewhere, and among the former into those overlapping with different classes of repeat elements. ↑ and ↓ indicate mutation types showing statistically significant differences relative to the genomic background SNV set (adjusted $p < 0.05$, two-sided Fisher's exact test). **b** Percentage contribution of COSMIC mutational signatures among different groups of SNVs; only signatures with nonzero values are shown. Asterisks indicate mutational signatures displaying >1% increase relative to the genomic background SNV set. Brief summaries of the aetiologies of affected signatures are shown on the right (descriptions taken from the COSMIC website). Source data are provided as a Source Data file.

such as Signatures 1, 3, 5, 6, 11, 12, 20, and 26. Such differences between different groups could be due to the heterogeneity of contributing mutational processes and redundancy among some COSMIC signatures.

It is important to note that COSMIC mutational signatures were designed for use with cancer genomes and so some germline mutational processes may not be well represented. Nevertheless, our analysis identified several candidate mutational processes associated with strong nucleosomes, such as the mutagenesis linked to DNA MMR (Signatures 6, 20, and 26) and DNA DSB repair (Signatures 3 and 5). Therefore, to gain deeper insights and to obtain independent evidence for these mutational processes, we examined multiple published genomic datasets below.

**MMR (Signatures 6, 20, and 26).** DNA MMR is a major pathway that is active during DNA replication: it mainly repairs mismatches and short INDELs introduced by DNA synthesis that have escaped polymerase proofreading. Mutations arising from inefficiencies in MMR are represented by Signatures 6, 20, and 26, which show increased contribution to de novo SNVs in the "all strong nucleosomes" group (2% increase collectively) and three repeat-based subgroups of mutations (1.6, 6.7, and 4.3% increase for "Alu", "L1", and "Others", respectively).

We analyzed somatic mutations from two sets of ultrahyper-mutated cancer genomes[40]. The first comprised genomes with driver mutations in the *POLE* gene encoding the catalytic subunit of DNA polymerase ε (Pol ε, the major replicase for the leading

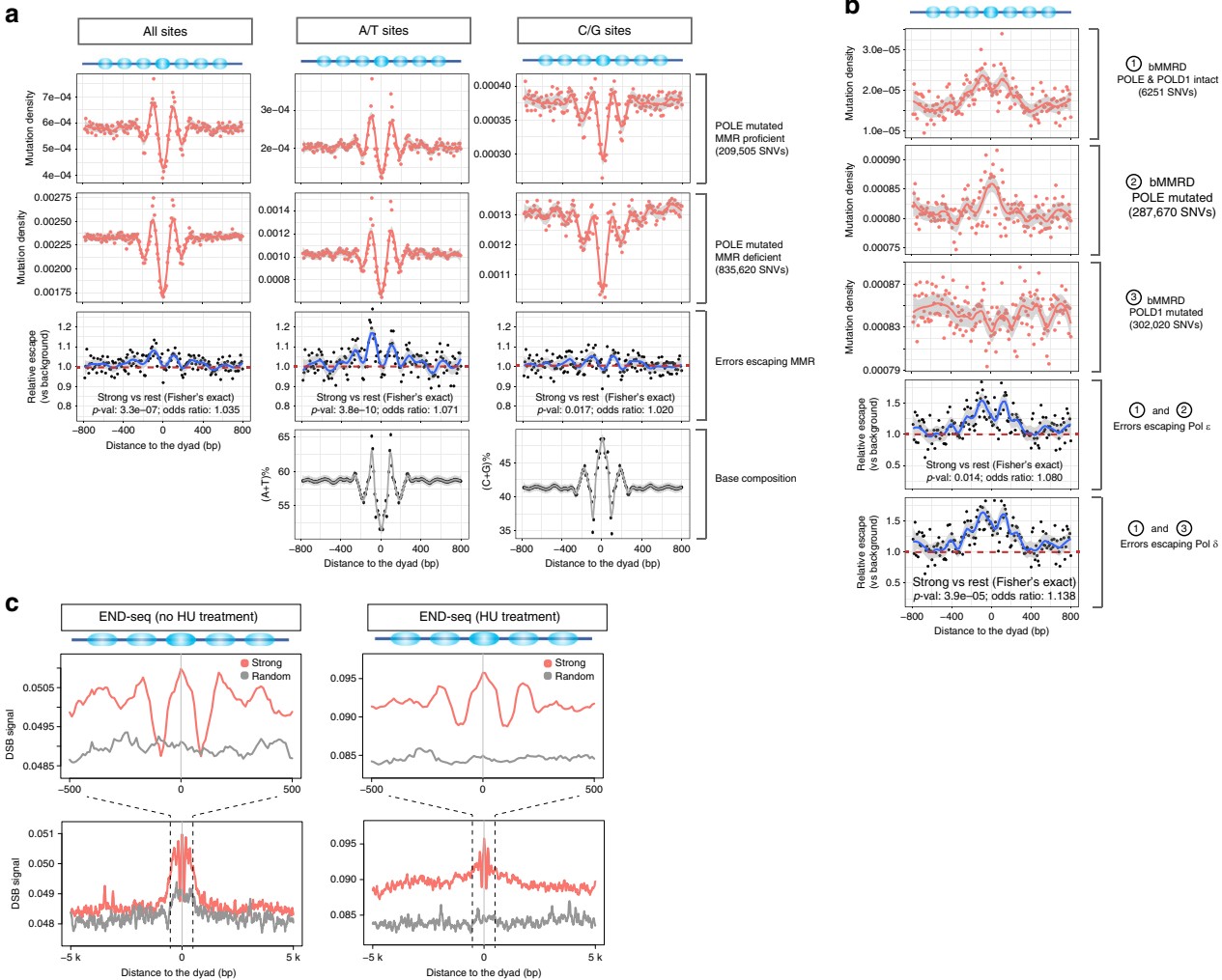

**Fig. 4 Mismatch repair (MMR), DNA polymerase fidelity, and double-strand breaks (DSB) explain increased mutation rates in strong nucleosomes.**
**a** Mutation density profiles relative to strong-nucleosome dyads in cancer genomes harboring driver mutations in the *POLE* and MMR pathway genes. Numbers of mutations used are indicated in the brackets. The MMR escape ratio compares the mutation densities in the MMR-proficient and MMR-deficient genomes. **b** Mutation density profiles relative to strong-nucleosome dyads for bMMRD cancer genomes with different driver mutation statuses in the *POLE* and *POLD1* genes. The escape ratios compare the mutation densities for Pol ε-deficient and Pol δ-deficient cancers with the proficient ones. **c** END-seq signal indicating the density of DSBs relative to strong-nucleosome dyads. HU hydroxyurea. Two-sided Fisher's exact test was used for testing the association of strong nucleosomal regions (dyad ± 95 bp) with differential MMR/polymerase performance. Source data are provided as a Source Data file.

strand) and in one or more of the core MMR genes (*MLH1*, *MSH2*, *MSH6*, *PMS1*, and *PMS2*). The second contained cancers with mutated *POLE* but intact MMR. As it is even more challenging to detect somatic mutations in tumor-derived data than resequencing of normal individuals, we focused this analysis on strong nucleosomes found in high-mappability regions of the genome (see Methods).

We reasoned that differences in mutation distributions between the two sets of genomes could be attributed to the MMR pathway. The overall mutation patterns are similar in both cases, with much higher mutation rates at strong-nucleosome boundaries and adjacent linker DNA than the surrounding regions (Fig. 4a). This implies that errors introduced during error-prone replication by a deficient Pol ε escape repair by the MMR pathway when they coincide with strong nucleosomes. Next, we calculated an "MMR escape ratio" to quantify the relative amount of replication errors that escapes MMR repair in the *POLE* only mutant cancers compared with the *POLE* and MMR double mutants. Strong nucleosomal regions (especially boundaries and adjacent linkers) display ~10% higher escape

ratios than the genome-wide background (Fig. 4a). A/T sites have higher escape ratios than C/G sites around strong nucleosomes. Despite different nucleotide frequencies, both C/G and A/T sites exhibit similarly elevated escape ratio profiles (dyads having lower values than linkers; Fig. 4a), suggesting that strong nucleosomes can contribute to the elevated escape ratios independent of sequence context. Moreover, the apparent ~200-bp periodicity in escape ratio and mutation density profiles are suggestive of associations with nucleosome positioning other than sequence context alone (Fig. 4a). Together, these observations strongly indicate a relationship between replication errors, MMR, and strong nucleosomes in elevating mutation rates.

**DNA polymerase fidelity (Signatures 10 and possibly 12).** We also studied the effect of strong nucleosomes on replication fidelity by examining data from children with inherited biallelic MMR deficiency (bMMRD)[41]; these include ultrahypermutated genomes arising from Pol ε and polymerase δ defects (Pol δ, the major replicase for the lagging strand; *POLD1* encodes the

catalytic subunit of Pol δ). We estimated Pol δ and Pol ε escape ratios (escaping the proofreading correction of polymerases) using the same reasoning as above (Fig. 4b). We found that strong nucleosomes have higher escape ratios for both polymerases relative to the genomic background (Fig. 4b), implying that they have lower replication fidelity in these regions. The proofreading escape ratios for both polymerases are even higher than that for MMR (Fig. 4a, b) and A/T sites display higher proofreading escape ratios than C/G sites (Supplementary Fig. 5a). Again, the periodic pattern in the relative escape profiles (Fig. 4b, Supplementary Fig. 5a) suggests that nucleosome positioning contributes to the heterogeneity in replicase fidelity across the genome.

The etiology of Signature 12 is currently unknown. Here, we found that it contributes 21.15–21.99% to mutations in *POLD1*-mutant bMMRD genomes (inferred by MutationalPatterns, Supplementary Fig. 5b), but much less for other bMMRD samples (0–2.88% for *POLE*-mutant, and 3.32–10.43% for *POLE/POLD1*-intact). This suggests that Signature 12 is probably associated with Pol δ and that many de novo mutations around strong nucleosomes arise from errors escaping Pol δ proofreading. Surprisingly, Signature 10, known to be associated with Pol ε deficiency, is absent from strong nucleosomal de novo SNVs (Fig. 3b). This suggests that although both Pol ε and Pol δ have high proofreading escape ratios (i.e., low fidelity) around strong nucleosomes (Fig. 4b), most of the replication errors that are eventually converted to de novo mutations are derived from lagging strand replicase Pol δ.

Reijns et al. showed that in budding yeast, Okazaki junctions formed during lagging strand replication tend to be near nucleosome dyads and display elevated mutation rates[14]. We tested this by reanalyzing the OK-seq data from human lymphoblastoid cells[42]. Unlike yeast, Okazaki junctions in humans are more frequently located in the linker regions (Supplementary Fig. 6) rather than the dyads, suggesting that the mutagenic effects of Okazaki junctions are different in the two organisms. This may partly be because yeast lacks the typical H1 histone found in human and other eukaryotes. However, the very short reads (single-ended 50 bp) of OK-seq data restricted our analysis to regions with high read mappability (covering ~10% of strong nucleosomes), limiting the strength of the conclusions here.

**DSBs (Signatures 3 and 5).** DSB repair represented by Signatures 3 and 5 is another potential mechanism involved in strong-nucleosome-associated mutations (Fig. 3b). Tubbs et al.[43] studied the genome-wide distribution of DSBs using END-seq and suggested that poly(dA:dT) tracts are recurrent sites of replication-associated DSBs. Our analysis of these data revealed a higher frequency of DSBs around strong nucleosomes compared with genomic background (Fig. 4c). The trend holds for experiments with and without hydroxyurea treatment (HU, a replicative stress-inducing agent), suggesting that strong nucleosomes are endogenous hotspots (i.e., without HU treatment) of DSBs during replication. It is notable that young Alu and L1 elements harbor prominent poly(dA:dT) tracts, which are enriched at the boundary and linker regions of strong nucleosomes (Supplementary Fig. 7a). The patterns of high DSB frequency still hold true when looking at strong nucleosomes associated with different repeats (Supplementary Fig. 7b, c). However, because the END-seq data were sequenced with single-ended 75 bp reads and majority of young Alu and L1 elements cannot be assessed with such short reads, we could not pursue further detailed analysis. We also note that because of the lack of END-seq data derived from naked DNA, it is difficult to assess the contribution of strong nucleosomes to the elevated DSB frequency independent

of the sequence context. Since DSB repair can be error-prone[44], even using high-fidelity homologous recombination, frequent DSB formation and subsequent error-prone repair likely contribute to the elevated mutation rates around strong nucleosomes.

**Strong nucleosomes and evolution of transposons.** Above, we highlighted that ~70% of strong nucleosomes are in Alu and L1 retrotransposons (Supplementary Fig. 1d). Upon examination of the subfamilies (Fig. 5a, b), we uncovered a strong enrichment for evolutionarily young L1s (e.g., L1PA2 to L1PA11) and Alus (e.g., AluY to AluSx). Since younger repeats have poorer mappability, strong nucleosomes are underdetected (Supplementary Fig. 8a) and these observations probably underestimate the true enrichment. This may also explain why several of the youngest L1 subfamilies (L1PA2 to L1PA5) have lower enrichments than the slightly older subfamilies (Fig. 5a, Supplementary Fig. 8a).

The preference for nucleosomes to occupy specific sections of Alu elements is supported by both in vitro and in vivo evidence[45–48]. We recapitulated these observations for strong nucleosomes using the Gaffney et al. MNase-seq data (Fig. 5c): there are two hotspots of strong nucleosomes in young Alus, which fade away in older elements. We also observed that younger Alus exhibit elevated de novo mutation rates compared with old ones (Fig. 5c). The weaker translational stability in older Alus is accompanied by reduced de novo mutation rates for both SNVs and INDELs (Fig. 5c). Thus, these data suggest an intriguing interplay between Alus, strong nucleosomes and mutation rates.

The histone octamer is thought to preferentially bind DNA sequences presenting lower deformation energy costs[49]. We estimated deformation energies using the nuScore software[49] based on the DNA sequence and nucleosome core particle structure. We found that Alus do indeed exhibit lower deformation energies than surrounding regions (Fig. 5c). Furthermore, the energies of Alu elements tend to increase with age, suggesting that the accumulated mutations in Alu sequences reduced their nucleosome-binding stability. This is also supported by comparing deformation energies of Alu consensus sequences (ancestral states) and those of current genomic sequences (Supplementary Fig. 8b). We further analyzed the 3′ end sequences of L1 elements harboring strong nucleosomes and observed similar patterns (Supplementary Fig. 8c, d).

## Discussion
Though the involvement of nucleosome organization in DNA damage and repair processes was recognized nearly 30 years ago[50], its genome-wide effects on germline mutation rates (particularly in higher eukaryotes) have remained poorly understood. Our analysis combining large-scale de novo mutation and nucleosome datasets in human provides several important insights into this topic.

A major finding is that strong translational positioning of nucleosomes is associated with elevated de novo mutation rates, which is also supported by observations using extremely rare variants in polymorphism data. The ability to use de novo mutations here allowed us to bypass confounding evolutionary factors such as selection, thus allowing direct assessment of the impact on mutation rates. Importantly, our statistical tests controlling for nucleosome occupancy and other related factors confirmed the significant contribution of translational stability to mutation rate variation and indicated that it affects not only strong nucleosomes but also other parts of the genome. Therefore, we have discovered a nucleosomal factor that significantly modulates germline mutation rate variation.

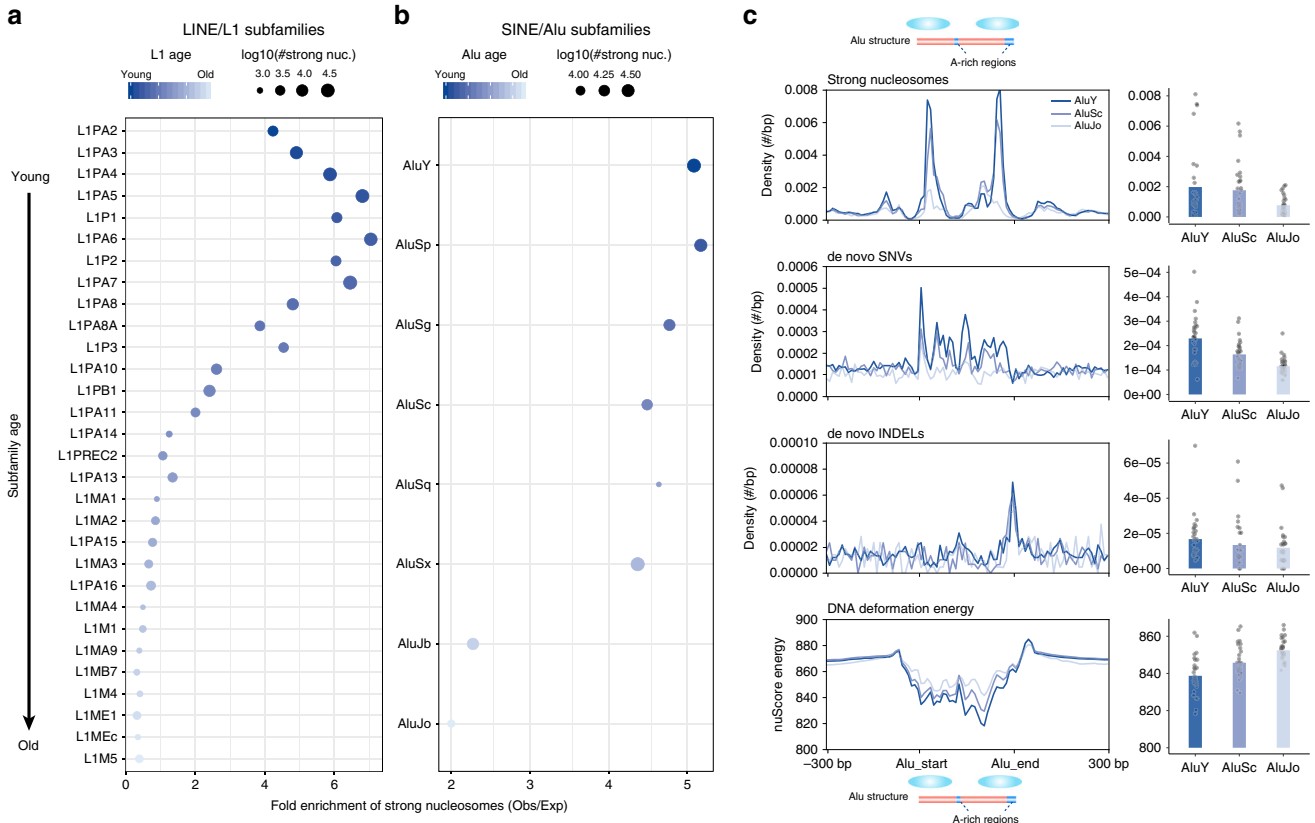

**Fig. 5 Strong nucleosomes are enriched for evolutionarily young LINE and SINE elements. a** Fold enrichment of strong-nucleosome occurrence in L1 subfamilies. The top 30 abundant subfamilies are shown ordered by evolutionary age. Dot sizes depict the numbers of strong nucleosomes and color-scale indicates the subfamily age. **b** Same as **a** but for Alu elements (only major subfamilies with age information presented here). **c** Densities of strong-nucleosome dyads, de novo SNVs, and de novo INDELs along the Alu sequences and flanking regions, grouped by Alu subfamilies of different ages. Bar plots show the average densities for all Alus of different subfamilies on the right, with dots representing the values of Alu bins in the left meta-profile panels. The bottom panel shows the average DNA deformation energies along Alu sequences estimated using nuScore. Profiles were plotted using Alu elements ≥250 bp and all elements were scaled up to a 300 bp region in the plots. Source data are provided as a Source Data file.

Investigating the underlying mutational processes responsible for this association remains challenging. Nevertheless, we obtained several informative results regarding potential mechanisms by leveraging published omics data related to DNA damage and repair. In doing so, we revealed that MMR, replicase fidelity, and DSB contribute to elevated mutation rates around strong nucleosomes. In particular, multiple sets of ultrahypermutated cancer data allowed us to quantify the performance of MMR and replicases by calculating the repair escape ratios. The results derived from analysis of cancer genomic data probably apply to germ cells because they agree nicely with the observations from our mutational signature analysis with de novo mutations. The precise molecular interactions determining the relationships between strong-nucleosome positioning, replicase fidelity, and DNA repair are still unclear. However, based on the evidence from our analysis with the omics data and previous studies[14,43,51], we speculate that strong nucleosomes may act as particularly strong barriers that impair the performance of the replication and repair machineries. There may be additional, unexamined effects on DNA damage and repair processes related to germline development, but many published genomic datasets about DNA damage and repair were generated in nongerm cells and with very short sequencing reads (e.g., <100 bp), which hinder accurate analysis. Improved sequencing strategies such as long-read sequencing and direct measurement in germ cells would benefit future related studies.

Interestingly, we found that strong nucleosomes are preferentially located within young LINE and SINE elements, two of the most common retrotransposons in the human and other mammalian genomes. Owing to their potentially deleterious effects, newly inserted retrotransposons are tightly repressed by multiple regulatory mechanisms, such as DNA methylation and H3K9me3 (ref. [52]). Strong-nucleosome positioning, which may mask access to the transcription machinery, could be another layer of the repressive system. Furthermore, the hypermutation in young SINEs/LINEs, partly contributed by associated strong nucleosomes, could lead to the rapid reduction of retrotransposition capacity. Therefore, the combination of strong-nucleosome positioning and hypermutation in SINEs/LINEs might have facilitated their expansion across the genome during evolution.

The decreasing numbers of strong nucleosomes in older LINE/SINE elements imply frequent nucleosome positioning changes during evolution. Since nucleosome positioning is strongly affected by the underlying DNA sequence, the decrease of positioning stability probably arises from the accumulation of mutations. A previous study suggested widespread selection for maintaining nucleosome positioning in the human genome[53]. Since a large majority of strong nucleosomes associated with SINE/LINE elements are expected to become nonstrong ones in future, selection for preserving positioning might not be as widespread as previously suggested, though it may happen at some particular regions or within a short evolutionary scale. Another evidence against strong selection for preserving positioning is that most genomic regions do not employ

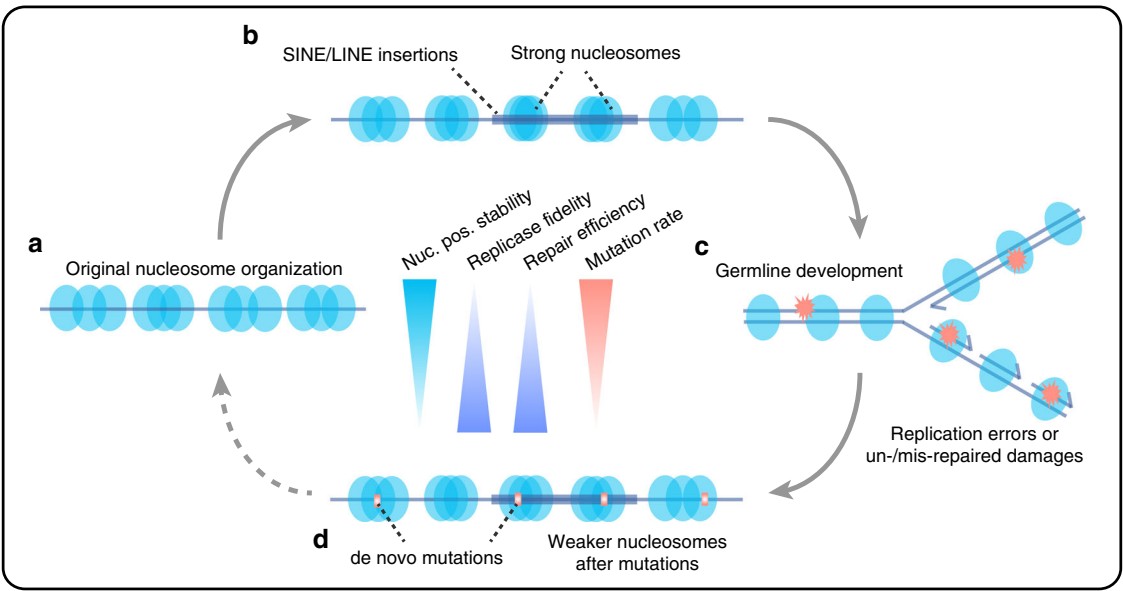

**Fig. 6 Proposed model of the interplay between nucleosome translational stability, mutation rate, and transposable elements. a** Most genomic regions are occupied by nucleosomes lacking strong translational stability. **b** Strong nucleosomes are preferentially associated with newly inserted SINE/LINE elements. **c** Strong nucleosomal regions are subject to high mutation rates during germline development, caused by mutational processes such as low replicase fidelity, inefficient MMR, and DSB repair. **d** Accumulation of mutations reduces translational stability of strong nucleosomes and reduces transposition capacity of transposable elements.

translationally stable positioning, possibly due to its relatively high mutagenic potential. Our data to some extent support the repositioning model proposed by Warnecke et al.[54].

Finally, we summarized our major findings in a proposed model in Fig. 6, which demonstrates the relationship among nucleosome positioning, mutation rate variation, retrotransposons, and evolution. Given the importance of germline de novo mutations in evolution and human diseases and the universal roles of nucleosomes in eukaryotic genome organization and regulation, our work should have profound implications in related research areas.

## Methods

**Mutation datasets**. De novo mutations identified in multiple large-scale trio sequencing project were downloaded from de novo-db (v1.6.1)[55]. Seven studies with >1000 de novo mutations[27–29,56–59] were considered in our analysis (Supplementary Fig. 1a). Extremely rare variants (derived allele frequency ≤ 0.0001) were obtained from Genome Aggregation Database (gnomAD, release 2.0.2)[30].

**Nucleosome datasets**. We used the 1,037,801 strong nucleosomes (i.e., translationally stable nucleosomes) identified based on MNase-seq data of sequenced seven lymphoblastoid cell lines from Gaffney et al.[32]. The original hg18-based coordinates of annotated nucleosomes were converted to hg19 using the "liftOver" tool from UCSC genome browser. The rotationally stable nucleosomes identified based on 49 DNase-seq samples (43 distinct cell types) were from Winter et al.[33]. We classified the human genome into three groups based on the nucleosome contexts (Fig. 1b): (i) regions covered by translationally stable ("strong") nucleosomes; (ii) regions covered by rotationally but not translationally stable nucleosomes; and (iii) the remaining genomic regions. Chromosomes X and Y were excluded from analysis as some other datasets used in our work lacked data for these chromosomes. As the nucleosome maps we used were not derived from germ cells, for downstream analysis we excluded the genomic regions in which nucleosome positioning were found to differ between human embryonic stem cells and differentiated fibroblasts[34]. Based on the positioning stability scores defined in Gaffney et al., we divided the one million strong nucleosomes into three categories of equal sizes with different levels of stability — "high", "middle", and "low", which were used for analysis shown in Fig. 1 and Supplementary Fig. 1.

**Accounting for mappability**. Sequencing read mappability can significantly affect variant calling results and other aligned read-depth based measurements (e.g., nucleosome occupancy). The sequencing reads for detecting de novo mutations used in our analysis were mainly 150 bp paired-end reads, with fragment sizes

ranging from 300–700 bp (Supplementary Fig. 1). We used the Genome Mappability Analyzer (GMA, v0.1.5)[36] to generate the mappability scores for simulated paired-end 150 reads with fragment sizes set to be 400 bp. Only the regions with GMA mappability scores of ≥90 (~2.59 Gb) were considered for most analyses, unless specified otherwise. We did not use the mappability tracks from ENCODE for the de novo mutation data, because those tracks were only for single-ended reads. For some analyses, additional filtering was applied if other associated datasets suffered from more severe mappability issues. For measuring nucleosome occupancy, we used the method described in the Gaffney et al. to simulate paired-end 25 bp reads matching the base compositions of MNase-seq data in the human genome, and then calculated per-base coverage depths by the simulated fragments. The 10 bp-bin ratios between the MNase-seq read coverage and the simulated read coverage were used for measuring the occupancy.

**De novo mutations in different nucleosome contexts**. We used Genomic Association Tester (GAT, v1.3.6)[60] to do the enrichment analysis, because it can perform simulations to estimate the expected numbers and calculate the statistical significance. We ran GAT by sampling 10,000 times (setting parameters "–ignore-segment-tracks –num-samples = 10,000 –num-threads = 5") to estimate the expected numbers of mutations in different contexts, which were then compared with the observed numbers. Low-mappability regions were excluded from analysis. A similar analysis was also done for the extremely rare variants of gnomAD. Analysis of meta-profiles along strong nucleosomes was done using deepTools (v3.1.3)[61].

**Statistical modeling of mutation rate variation**. As described in the main text, for a given genomic position, we defined two variables (see Eqs. (1) and (2) below) regarding the translational positioning of nearby nucleosomes (Fig. 2a)

$$d_{\text{mean}} = \frac{\sum_{i=1}^{n} d_i}{n}, 0 \le d \le 95, \tag{1}$$

$$d_{\text{var}} = \frac{\sum_{i=1}^{n} (d_i - d_{\text{mean}})^2}{n-1}, \tag{2}$$

where $d$ is the distance between a MNase-seq midpoint to the focal site. We considered MNase-seq midpoints within ±95 bp of the focal site, because genome-wide nucleosome repeat length was estimated to be 191.4 bp for the Gaffney et al. data[32]. Genomic sites having ≥10 MNase-seq midpoints within ±95 bp were considered from analysis (covering 2.49 Gb of usable sites). The measurements for nucleosome occupancy were 10 bp-bin ratios between the MNase-seq read coverage and the simulated read coverage. We did not use the positioning score $S(i)$ defined in Gaffney et al. to measure positioning stability in our modeling analysis, because $S(i)$ was designed for identifying the stable dyads and so for non-dyad positions it does not represent the positioning stability properly.

RNA expression, DNA methylation, and chromatin accessibility (ATAC-seq) data from human spermatogonial stem cells were from Guo et al.[62]. For the RNA-seq and ATAC-seq data from Guo et al., because the genome-wide read signal tracks were not available, we downloaded, processed, and mapped the raw reads to generate the genome-wide tracks. Since suitable data for histone modifications in human germ cells were not available, we used the ChIP-seq data of human embryonic stem cells from ENCODE[63]. Replication timing data (Repli-seq of GM12878) were also from ENCODE. The data of recombination rates were from the HapMap project[64].

A binary logistic regression framework was used to assess the contribution of different factors to mutation rate variation across the genome systematically. The logistic regression model is described as Eqs. (3) and (4) below

$$\mu = \Pr(y = 1) = \frac{\exp\left(\beta_0 + \beta_1 X_1 + \cdots + \beta_p X_p\right)}{1 + \exp\left(\beta_0 + \beta_1 X_1 + \cdots + \beta_p X_p\right)} = \frac{\exp(\mathbf{X\beta})}{1 + \exp(\mathbf{X\beta})}, \quad (3)$$

$$\text{logit}(\mu) = \log\left(\frac{\mu}{1 - \mu}\right) = \mathbf{X\beta}, \quad (4)$$

where $\mu = \Pr(y = 1)$ denotes the probability that a genomic position is mutated (for testing individual SNV mutation types, e.g., A > T, $\mu$ is the probability that a site is mutated to a specific nucleotide), $\mathbf{X}$ represents the observations for the considered variables (categorical or continuous, e.g., $d_{mean}$, $d_{var}$, adjacent nucleotides, etc.), and $\mathbf{\beta}$ is the vector of parameters to be estimated.

We used the Bayesian logistic regression model implemented in the "bayesglm"[65] of the R package "arm" (v1.10.1), which was reported to perform well in handling the complete separation issue in logistic regression models[65]. The setting of priors in "bayesglm" was "prior.scale = 2.5, prior.df = 1". The complete separation issue is common when one class is rare relative to the other and (or) there are many regressors in a model. As we had only ~300,000 de novo mutations, the probability for a given site to be mutated in our data is ~1/10,000, which is a rare event.

Within the logistic regression framework, we compared the full model with all considered variables to a reduced model without one specific variable by performing likelihood-ratio tests in R ("anova" function) to assess the significance for each variable. The resulting p values of a set of likelihood-ratio tests were adjusted for multiple testing with Benjamini–Hochberg correction. The R package "pscl" (v1.5.2) was used to calculate the McFadden's pseudo $R^2$ of the regression models.

To perform the regression analysis, we generated the data of all variables for the de novo mutation sites and subsampled a fraction of the nonmutated sites as the control sites. We did not use all the nonmutated sites in the genome as it would lead to a large imbalance in the sizes of two classes ("mutated" and "nonmutated") and much larger computational burden. For de novo SNVs, we randomly generated 2,490,967 nonmutated sites (about 1/1000 of the accessible genome, about ten times as many as de novo SNVs), and 249,669 nonmutated sites (about 1/10,000 of the accessible genome, about ten times as many as de novo INDELs) for INDELs. For de novo INDELs, we used the INDELs of ≤5 bp for regression analysis, because long INDELs were rare and may have high false positive/negative rates. For RNA expression, DNA methylation, chromatin accessibility, replication timing, recombination rate, and histone modification data, we used the average value of the ±10 bp of a focal site for each specific feature based on the genome-wide signal tracks. We also assessed different window sizes (±5 and ±20 bp), which led to similar results.

For SNVs, we performed logistic regression tests for mutation types at A/T sites and C/G sites separately and distinguished C/G sites in CpG and non-CpG contexts. We also tested for nine individual SNV mutation types (three for A/T sites, three for C/G sites at CpG contexts, and three for non-CpG contexts, Supplementary Fig. 3). The regression coefficients for the full model of each test are given in Supplementary Data 1.

Since the variable $d_{mean}$ has a nonmonotonic relationship with mutation rates, we binned the values into five categories: [0,18], [19, 36], [37, 54], [55, 73], and [74, 95] (first four bins implying nucleosome-bound regions, and the last bin implying close to the linker). We applied log transformation to the variables $d_{var}$ and "expression level", because the log transformation can largely improve statistical significance of these two variables.

In the regression models mentioned above, we did not consider the nonadditive effects of adjacent nucleotides (±5 bp). When we tried adding nonadditive effects for ±5 nucleotides (considering only two-way interactions; taking a much longer running time), we got similar results regarding the association of translational stability ($d_{var}$) and mutation rates (Supplementary Fig. 4). We also tried using the 7-mer mutability estimates from Carlson et al.[31], which incorporated nonadditive effects among ±3 nucleotides, as predictors in the regression models.

To evaluate how the repeat status affects the effects of translational stability on mutation rates, we added the repeat status ("Alu", "L1", "other repeat" or "nonrepeat") as a predictor to regression models, and also ran the regression tests for different repeat/nonrepeat regions separately.

**Analysis of mutational processes.** COSMIC mutational signatures are based on frequencies of mutations in tri-nucleotide contexts. Since the regions associated with strong nucleosomes have different tri-nucleotide composition relative to genome background, we first normalized the mutation type frequencies in regions associated with strong nucleosomes as this: set $F_{i,strong}$ for the occurrence of a specific mutation type (e.g., T[T > C]T), $N_{i,strong}$ for the occurrence of the considered tri-nucleotide context (e.g., TTT) in strong-nucleosome regions, and $N_{i,genome}$ for the occurrence of the considered tri-nucleotide context in the whole-genome background, then the corrected occurrence of a mutation type for strong nucleosomes is $N'_{i,strong} = F_{i,strong} \div N_{i,strong} \times N_{i,genome}$. Two-sided Fisher's exact tests were performed to identify mutation types that show significant increase or decrease in strong-nucleosome regions relative to genome background. The contingency table used for running "fisher.test" in R for a specific mutation type is $matrix\ (c(F_{i,strong},\ N_{i,strong} - F_{i,strong},\ F_{i,genome} - F_{i,strong},\ (N_{i,genome} - N_{i,strong}) - (F_{i,genome} - F_{i,strong})),\ ncol = 2)$, where $F_{i,strong}$ and $F_{i,genome}$ are the occurrences of the considered mutation type and $N_{i,strong}$ and $N_{i,genome}$ for the occurrences of the considered tri-nucleotide context. Benjamini–Hochberg method was used for multiple testing correction.

The contribution of COSMIC mutational signatures[37] to different sets of mutations (de novo SNVs and somatic mutations from bMMRD samples) was predicted using the "fit_to_signatures" function in the R package "MutationalPatterns" (v1.8.0)[38]. For the sets of de novo SNVs associated with strong nucleosomes, the corrected frequencies described above were used for running "fit_to_signatures."

Mutations in *POLE* in cancers can lead to reduced base selectivity and/or deficient proofreading during replication, producing unusually large numbers of mutations (so called "ultrahypermutation"), which facilitated our analysis. *POLE* mutated genomes from PCAWG project[40] were used to evaluate the differential MMR efficiency between strong and nonstrong-nucleosome regions. We compared the mutation densities in cancer genomes with *POLE* mutated and a deficient MMR (four individual samples) to those with *POLE* mutated and a proficient MMR (six samples). The MMR pathway was considered deficient if a driver mutation (annotated by the PCAWG consortium) was found in one of five MMR core genes — *MLH1*, *MSH2*, *MSH6*, *PMS1*, and *PMS2*.

For a given bin (10 bp size) in the meta-profile, we calculated the relative MMR escape ratio relative to genomic background around strong nucleosomes as described in the following equation:

$$R_i^{escape} = \frac{\frac{m_i^{POLE^*, MMR^{WT}}}{m_i^{POLE^*, MMR^*}}}{\frac{\overline{m}^{POLE^*, MMR^{WT}}}{\overline{m}^{POLE^*, MMR^*}}}, \quad (5)$$

where $m_i$ is the mutation density for the $i$th bin (observed number of mutations in the $i$th bin divided by the bin size), and $\overline{m}$ is the genome-wide average mutation density of a specific sample group (observed number of mutations in the simulated windows divided by the total window size), estimated by simulating random windows in the genome. "*" and "WT" depict mutant and wildtype, respectively. A similar logic was used when evaluating relative proofreading escape ratios of Pol ε (mutated *POLE*) and Pol δ (mutated *POLD1*) using the somatic mutation data from the bMMRD project[41].

When analyzing PCAWG and bMMRD data, to account for potential mappability issues, we focused on the highly mappable regions based on the CrgMapability scores from ENCODE. We used CrgMapability scores here, which are more stringent than GMA ones, because detecting somatic mutations in tumors is more difficult than for ordinary individual resequencing data. We considered the strong nucleosomes that have a 100-mer CrgMapability score of 1 (meaning any 100-bp read from these regions can be mapped uniquely in the genome) within ±800 bp of the dyads. We then simulated a same number of 1600 bp-sized regions from the genome that satisfy the mappability requirement to calculate the background mutation density. Note that in theory the mappability issue in the relative escape ratios should be very small because the two sets of samples have the same mappability for a given bin and the ratio calculation in Eq. (5) normalizes the effects of different mappability among regions.

Two-sided Fisher's exact tests were performed to test the association of strong nucleosomal regions (dyad ± 95 bp) with differential MMR/polymerase performance. For example, for testing the MMR performance, the contingency table used for running "fisher.test" in R is $matrix\left(c\left(N_{strong}^P, N_{all}^P - N_{strong}^P, N_{strong}^d, N_{all}^d - N_{strong}^d\right)\right), ncol = 2)$, where $N_{strong}^P$ and $N_{all}^P$ are the numbers of mutations located in strong nucleosomal regions (dyad ± 95 bp) and all the considered regions for the MMR-proficient sample respectively, and $N_{strong}^d$ and $N_{all}^d$ for the MMR-deficient sample. The same method was used to test the performance of Pol ε and Pol δ around strong nucleosomes.

The raw reads of OK-seq data[42] were downloaded from NCBI and mapped to the human genome. We kept only the uniquely mapped reads for inferring Okazaki junctions. The very 5′ end sites of aligned reads (separating reads mapped to Watson and Crick strands) were considered putative Okazaki junction signals.

To investigate DSBs around strong nucleosomes, we downloaded the genome-wide tracks of human END-seq data (GSM3227951 and GSM3227952)[43]. Because the reads of END-seq data were single-ended 75 bp, we considered the strong nucleosomes that have a 75-mer CrgMapability score of 1 within ±500 bp of the strong-nucleosome dyads for analysis.

**Strong nucleosomes in different repeat contexts**. GAT[60] was used to estimate the expected numbers of strong nucleosomes in different contexts (sampling ≥ 1000 times), which were compared with the observed numbers. The annotations of repeat elements (February 2009, Repeat Library 20140131) were downloaded from RepeatMasker website[66]. We also did GAT analysis for LINE/L1 and SINE/Alu subfamilies of different ages. The age information of repeat families was from Giordano et al.[67]. For generating the MNase-seq midpoints along the repeat consensus sequences, we made use of the alignment information in the Repeat-Masker result files ("hg19.fa.align.gz") and mapped the hg19-based coordinates to the coordinates in the consensus sequences. Strong nucleosomes appear to be underdetected in very young L1 elements, which we think is due to difficulties in mapping short MNase-seq reads (Alus are easier to map because they are much smaller).

Nucleosome deformation energies of all sites in the human genome were estimated using nuScore (v1.0)[49]. We also ran nuScore to estimate the deformation energies of Alu and L1 subfamilies consensus sequences. For the L1 analysis shown in Supplementary Fig. 8, we only considered the 3′ end regions of L1 subfamilies, because 5′ end regions of L1 elements are usually truncated in the genome and their subfamily identities are difficult to be determined.

**Reporting summary**. Further information on research design is available in the Nature Research Reporting Summary linked to this article.

## Data availability

All the analyses in this study were based on published datasets. Links for main published datasets used in the study: MNase-seq data[32] [http://eqtl.uchicago.edu/nucleosomes/mnase_seq.html]; de novo-db[55] [http://denovo-db.gs.washington.edu/denovo-db/]; gnomAD data[30] [http://gnomad.broadinstitute.org/]; repeat annotations[66] [http://www.repeatmasker.org/species/hg.html]; PCAWG data[40] [https://dcc.icgc.org/pcawg/]; bMMRD data[41] [https://www.ebi.ac.uk/ega/studies/EGAS00001001112]; END-seq data[43] GSE116321. Other data generated in the study are available from the corresponding author on reasonable request. The source data underlying Figs. 1c–f, 2b–d, 3a, b, 4a–c, and 5a–c and Supplementary Figs. 1b–e, 2, 3, 4a–d, 5a, b, 6b, 7a–c, and 8a–d are provided as a Source Data file.

## Code availability

Custom scripts and associated input data are available at the ZENODO repository; DOI: 10.5281/zenodo.3598517 [https://doi.org/10.5281/zenodo.3598517].

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

## Acknowledgements

We are grateful to Tobias Warnecke, John Diffley, Anob Chakrabarti, and Sara Rohban for insightful comments. We thank Peter Van Loo, Jonas Demeulemeester, and Maxime Tarabichi for assistance in accessing the PCAWG genomic data. We appreciate obtaining access to the de novo mutation data on SFARI Base. This work is supported by the Francis Crick Institute that receives its core funding from Cancer Research UK (FC001110), the UK Medical Research Council (FC001110), and the Wellcome Trust (FC001110) (N.M.L.). N.M.L. is also supported by a Wellcome Trust Investigator Award and core funding from the Okinawa Institute of Science & Technology. C.L. is funded by an EMBO long-term postdoctoral fellowship (ALTF 1499–2016).

## Author contributions

C.L. conceived the project, performed the analyses, and drafted the manuscript; N.M.L. supervised the project and co-wrote the manuscript.

## Competing interests

The authors declare no competing interests.
