## [Peer Review File · Nature Communications]

Reviewers' comments:

Reviewer #1 (Remarks to the Author):

The authors have done a good job responding to the reviews, and I have no further substantial comments. I just want to comment on their response to my comment 1.6: They say it isn't possible to agnostically infer germline mutational signatures without sequencing data from multiple germline samples each experiencing different combinations of mutational processes, and I was confused by this because there exist thousands of human germline DNA sequences at this point. Perhaps they mean that these individuals are much more similar to each other than cancer genomes tend to be, but the right analysis to do in the context of this paper would be to consider different genomic subsets to essentially be different individuals (e.g. look at regions bound by strong nucleosomes separately from regions with weak or absent nucleosomes, and/or separate out LINEs and SINEs from non repetitive DNA) and learn agnostically which mutational signatures can best explain the set of differences among these compartments. I don't think it's essential for them to do this analysis, as long as they acknowledge that cancer signatures are imperfect proxies, but wanted to point out how the agnostic inference would be possible.

Reviewer #2 (Remarks to the Author):

The authors have adequately addressed most of my major concerns, especially the one regarding mutational signature analysis. However, I'm still not fully convinced that the analysis of transposable elements is sufficient to "resolve a debate about selective pressure on nucleosome-positioning". Below I outline my reasoning. I hope the authors could consider radically reduce or revise section 2.5.

Major concern regarding the re-positioning model:

Again I wanted to point out that the various models all aim to explain one phenomenon: the observed uneven sequence divergence in and around nucleosomes, with substitution rates seemingly reduced for certain based changes (e.g. strong-to-weak) but elevated for others (e.g. weak-to-strong). The two traditional models are biased mutation and biased selection in DNA sequence bound to nucleosome, whereas the model proposed by Warnecke et al (2013) suggests that the uneven divergence patterns could also arise, if there is nucleosome re-positioning following sequence substitution, because certain nucleotide contexts tend to attract (or repel) nucleosome. In another words, Warnecke's model is that mutation causes nucleosome re-positioning rather than nucleosome presence causes biased mutation.

Because this manuscript is primarily focused on the effect of translationally stable nucleosomes on the mutation rate (i.e., nucleosome stability causes elevated mutation rate), I strongly feel that most results presented (up to line 425) are irrelevant to Warnecke's model. Instead, the findings actually support the older biased mutation model.

To support Warnecke's model, one needs to demonstrate that the nucleosome positions shift after sequence substitution. To this end, the comparison of deformation energies of current and consensus sequences (Fig S8) provides the only piece of support. However, the evidence is still insufficient, because in order to explain the highly regular divergence patterns around nucleosomes, Warnecke's model requires nucleosomes to shift to new positions in a predictable manner rather than just wobble around its ancestral position. In other words, in Warnecke's model, nucleosomes are frequently repositioned to new locations with the greatest affinity, and the overall stability may to decrease, increase, or stay the same. However, in this manuscript, the authors present a model where the nucleosome stability always decreases with time – then there is a conundrum: how do strong nucleosomes form in the first place (i.e., in Fig 6, how does state a transit to state b)?

In addition, the loss of strong nucleosomes with time, if true, only suggests that selection is not strong enough to completely eliminate random mutations, but it does not argue for complete

absence of selection. If there is relatively weak selective pressure maintaining nucleosome positions and the underlying sequences, mutations will still accumulate, despite at a rate lower than what's expected under neutral evolution. Therefore, the results presented here argue against very strong selection in transposable elements but do not fully exclude the possibility of weaker selection.

A final concern about the analysis presented in Fig 5 is how representative are transposable elements of the other genomic regions. Is the uneven divergence pattern around nucleosomes also observed in transposable elements? Are mutagenic effects of strong nucleosomes similar in and outside transposable elements? Are transposable elements subject to similar selection pressure as genic or regulatory regions? Without knowing the answers, it seems premature to conclude that "Depletion of strong nucleosomes in older transposons suggests frequent re-positioning during evolution, thus resolving a debate about selective pressure on nucleosome-positioning".

Minor points:

The authors conclude that nucleosome stability has a significant effect on mutation rate by showing that including *dvar*, a metric for translational stability, significantly improves the pseudo R² in the regression model. In the model presented in Fig 2, all investigated factors are assumed to have linear, additive effects on mutation rate, with no interaction effects. The authors briefly investigated the effects of more complex sequence contexts by including two-way interaction between +/-5bp adjacent nucleotide or including 7-mer mutability estimate from Carlson et al (2018) (Fig 4a). However, there could exist more complex interaction (e.g., three-way interaction between adjacent nucleotides, interaction between histone modification state and nucleotide type) that can explain additional variation in the mutation rate and thus reduce the effect of translational stability. Understandably, it's impossible to test all possible models with non-linear or interaction effects, but it should be clearly noted that the logistic regression model has its limitations.

It should be explicitly mentioned that the expectation presented in Fig 1 does not take into account sequence composition or other genomic features.

In lines 151-152, it is claimed that "Therefore, the above observations probably underestimate the true enrichment of de novo mutations in strong nucleosomes". However, in regions with low mappability, there is likely under-detection of both de novo mutations and strong nucleosomes, so how do we know whether the enrichment of mutations in strong nucleosomes is over- or underestimated?

A recent reference, Halldorsson et al. (2019), should be added to support the effect of recombination on mutation rate (page 3, line 44).

In lines 513-515, the authors state that "The results probably apply to germ cells because i) they agree nicely with the observations from our mutational signature analysis with de novo mutations". What results specifically do they refer to?

In lines 515-517, it is stated that "ii) recent studies suggested that replicative errors account for majority of mutations arising in both somatic and germ cells (Tomasetti and Vogelstein 2015; Tomasetti et al. 2017)". However, there is still debate regarding the contribution of replication-error-induced mutations to germline mutations (see Gao et al 2019 for a counter-example). Even in the cancer field, most people believe that unrepaired DNA damage, in addition to replication errors, plays an important role in mutagenesis. In fact, the author's own mutational signature analysis suggests that Signature 1, presumably due to cytosine deamination, contributes a large fraction (>25%) of de novo mutations (Fig 3b). Therefore, the sentence mentioned above is incorrect and need to be revised. More generally, there is currently no consensus whether somatic and germline mutations are caused by similar mutational processes (and whether the same process has similar relative contributions).

Point-by-point responses (in BLUE) to referees' comments (in BLACK)

1. Referee #1's comment

The authors have done a good job responding to the reviews, and I have no further substantial comments. I just want to comment on their response to my comment 1.6: They say it isn't possible to agnostically infer germline mutational signatures without sequencing data from multiple germline samples each experiencing different combinations of mutational processes, and I was confused by this because there exist thousands of human germline DNA sequences at this point. Perhaps they mean that these individuals are much more similar to each other than cancer genomes tend to be, but the right analysis to do in the context of this paper would be to consider different genomic subsets to essentially be different individuals (e.g. look at regions bound by strong nucleosomes separately from regions with weak or absent nucleosomes, and/or separate out LINEs and SINEs from non repetitive DNA) and learn agnostically which mutational signatures can best explain the set of differences among these compartments. I don't think it's essential for them to do this analysis, as long as they acknowledge that cancer signatures are imperfect proxies, but wanted to point out how the agnostic inference would be possible.

Author's response:

We thank the referee for the explanation. We apologize for misunderstanding the referee's original comment and we agree that it is possible to perform the analysis as described above. We have not pursued this further as it isn't essential for the paper; however we have pointed out the limitations of using cancer signatures in lines 294-296 "It is important to note that COSMIC mutational signatures were designed for use with cancer genomes and so some germline mutational processes may not be well represented."

2. Referee #2's comments

2.1 Referee's comment:

The authors have adequately addressed most of my major concerns, especially the one regarding mutational signature analysis. However, I'm still not fully convinced that the analysis of transposable elements is sufficient to "resolve a debate about selective pressure on nucleosome-positioning". Below I outline my reasoning. I hope the authors could consider radically reduce or revise section 2.5.

Author's response:

We thank the referee for the careful evaluation and discussion. After considering the referee's comments, we have substantially revised section 2.5 and the related content. We provide a detailed response below.

2.2 Referee's comment:

Major concern regarding the re-positioning model:

Again I wanted to point out that the various models all aim to explain one phenomenon: the observed uneven sequence divergence in and around nucleosomes, with substitution rates seemingly reduced for certain based changes (e.g. strong-to-weak) but elevated for others (e.g. weak-to-strong). The two traditional models are biased mutation and biased selection in DNA sequence bound to nucleosome, whereas the model proposed by Warnecke et al (2013) suggests that the uneven divergence patterns could also arise, if there is nucleosome re-positioning following sequence substitution, because certain nucleotide contexts tend to attract (or repel) nucleosome. In another words, Warnecke's model is that mutation causes nucleosome re-positioning rather than nucleosome presence causes biased mutation.

Because this manuscript is primarily focused on the effect of translationally stable nucleosomes on the mutation rate (i.e., nucleosome stability causes elevated mutation rate), I strongly feel that most results presented (up to line 425) are irrelevant to Warnecke's model. Instead, the findings actually support the older biased mutation model.

To support Warnecke's model, one needs to demonstrate that the nucleosome positions shift after sequence substitution. To this end, the comparison of deformation energies of current and consensus sequences (Fig S8) provides the only piece of support. However, the evidence is still insufficient, because in order to explain the highly regular divergence patterns around nucleosomes, Warnecke's model requires nucleosomes to shift to new positions in a predictable manner rather than just wobble around its ancestral position. In other words, in Warnecke's model, nucleosomes are frequently repositioned to new locations with the greatest affinity, and the overall stability may to decrease, increase, or stay the same. However, in this manuscript, the authors present a model where the nucleosome stability always decreases with time – then there is a conundrum: how do strong nucleosomes form in the first place (i.e., in Fig 6, how does state a transit to state b)?

In addition, the loss of strong nucleosomes with time, if true, only suggests that selection is not strong enough to completely eliminate random mutations, but it does not argue for complete absence of selection. If there is relatively weak selective pressure maintaining nucleosome positions and the underlying sequences, mutations will still accumulate, despite at a rate lower than what's expected under neutral evolution. Therefore, the results presented here argue against very strong selection in transposable elements but do not fully exclude the possibility of weaker selection.

A final concern about the analysis presented in Fig 5 is how representative are transposable elements of the other genomic regions. Is the uneven divergence pattern around nucleosomes also observed in transposable elements? Are mutagenic effects of strong nucleosomes similar in and outside transposable elements? Are transposable elements subject to similar selection pressure as genic or regulatory regions? Without knowing the answers, it seems premature to conclude that "Depletion of strong nucleosomes in older transposons suggests frequent re-positioning during evolution, thus resolving a debate about selective pressure on nucleosome-positioning".

Author's response:

We thank the referee for the comments.

We agree that the presented data in this work are not sufficient to fully support Warnecke's re-positioning model and resolve the debate regarding the uneven

divergence patterns around nucleosomes. We have removed and revised the text related to Fig. 5 to resolve these concerns. More specifically, we completely removed the last two paragraphs in section 2.5 and revised the related paragraph in the Discussion section. We also removed the sentence “thus resolving a debate about selective pressure on nucleosome-positioning” from the Abstract. We also agree that our data do not exclude the possibility of weaker selection for preserving positioning. We acknowledge this in the revised Discussion section. Note that all of these will not affect the proposed model in Fig. 6, which doesn’t mention the re-positioning model.

Regarding the concern “the authors present a model where the nucleosome stability always decreases with time – then there is a conundrum: how do strong nucleosomes form in the first place (i.e., in Fig 6, how does state a transit to state b)?”, our model describes general patterns in the genome but does not rule out the possibility that some regions maintain high nucleosome stability (at least on a short time scale). It is also possible that random mutations could increase the translational stability in some regions. This is analogous to CpGs in transposable elements: a large fraction of the CpGs in newly inserted transposable elements tend to be lost over time, but some transposable elements are capable of transposing to other regions before losing their CpGs. Likewise, a fraction of transposable elements with sequences attracting strong nucleosomes could proliferate in the genome, though most (not all) of these elements tend to lose the strong nucleosomes over time.

Author’s action:

a) The following two paragraphs in the Section 2.5 have been removed:

“Studies have suggested that natural selection appears to preserve nucleosome positioning during evolution ... The use of de novo mutations helps resolve this debate to some extent.

As we showed above, there is considerable de novo mutation rate variation around strong nucleosomes ... it may happen at some particular regions or within a short evolutionary scale.”

b) The updated paragraph in the Discussion section:

“The decreasing numbers of strong nucleosomes in older LINE/SINE elements imply frequent nucleosome positioning changes during evolution. Since nucleosome positioning is strongly affected by the underlying DNA sequence, the decrease of positioning stability probably arises from the accumulation of mutations. A previous study suggested widespread selection for maintaining nucleosome positioning in the human genome⁵³. Since a large majority of strong nucleosomes associated with SINE/LINE elements are expected to become non-strong ones in future, selection for preserving positioning might not be as widespread as previously suggested, though it may happen at some particular regions or within a short evolutionary scale. Another evidence against strong selection for preserving positioning is that most genomic regions do not employ translationally stable positioning, possibly due to its relatively high mutagenic potential. Our data to some extent support the re-positioning model proposed by Warnecke et al.⁵⁴.”

c) Updated sentence in the Abstract:

“Depletion of strong nucleosomes in older transposons suggests frequent positioning changes during evolution.”

2.3 Referee’s comment:

Minor points:

The authors conclude that nucleosome stability has a significant effect on mutation rate by showing that including *dvar*, a metric for translational stability, significantly improves the pseudo R² in the regression model. In the model presented in Fig 2, all investigated factors are assumed to have linear, additive effects on mutation rate, with no interaction effects. The authors briefly investigated the effects of more complex sequence contexts by including two-way interaction between +/-5bp adjacent nucleotide or including 7-mer mutability estimate from Carlson et al (2018) (Fig 4a). However, there could exist more complex interaction (e.g., three-way interaction between adjacent nucleotides, interaction between histone modification state and nucleotide type) that can explain additional variation in the mutation rate and thus reduce the effect of translational stability. Understandably, it’s impossible to test all possible models with non-linear or interaction effects, but it should be clearly noted that the logistic regression model has its limitations.

Author’s response:

We thank the referee for this comment. We now acknowledge the limitations of the logistic regression model in the revised manuscript.

Author’s action:

We updated the following sentences in section 2.3:

“We acknowledge the limitation that logistic regression model cannot assess all higher-order interactions among the long stretches of nucleotides which guide nucleosome positioning. It is also impossible to evaluate all possible interactions between local sequences and many functional features. Nonetheless, we achieved similar statistical significance ...”

2.4 Referee’s comment:

It should be explicitly mentioned that the expectation presented in Fig 1 does not take into account sequence composition or other genomic features.

Author’s response:

We thank the referee for the comment. We have clarified this in the revised manuscript.

Author’s action:

We modified the first sentence referring to the Fig. 1c and Fig. 1d:

“Genomic regions containing strong nucleosomes have ~30% more de novo SNVs (Fig. 1c) and ~15% more de novo INDELs (Fig. 1d) than expected (without considering the sequence composition and other genomic features).”

2.5 Referee's comment:

In lines 151-152, it is claimed that “Therefore, the above observations probably underestimate the true enrichment of de novo mutations in strong nucleosomes”. However, in regions with low mappability, there is likely under-detection of both de novo mutations and strong nucleosomes, so how do we know whether the enrichment of mutations in strong nucleosomes is over- or under-estimated?

Author's response:

We thank the referee for this comment. Here, we made the inference mainly based on the data and analysis in the regions of previously annotated nucleosomes. Among the considered regions, those harbouring strong nucleosomes extensively overlap repeats, which likely have more un-detected de novo mutations. Therefore, in this context the enrichment of de novo mutations in strong nucleosomes is likely underestimated.

It is true that we don't know the actual patterns for the low-mappability regions in which nucleosomes (and many de novo mutations) could not be annotated. In the revised manuscript, we toned this down by changing “probably” to “may” in the sentence.

Author's action:

Updated sentence:

“Therefore, the above observations may underestimate the true enrichment of de novo mutations in strong nucleosomes.”

2.6 Referee's comment:

A recent reference, Halldorsson et al. (2019), should be added to support the effect of recombination on mutation rate (page 3, line 44).

Author's response:

We thank the referee for the comment. We have added this reference in the revised manuscript.

Author's action:

Updated sentence (ref #9 is Halldorsson et al. (2019)):

“Studies revealed factors linked to local mutation rate variation, including sequence context⁵, replication timing⁶, recombination rate⁷⁻⁹, DNA accessibility¹⁰ and histone modifications^{5,11}.”

2.7 Referee's comment:

In lines 513-515, the authors state that “The results probably apply to germ cells because i) they agree nicely with the observations from our mutational signature analysis with de novo mutations”. What results specifically do they refer to?

Author's response:

We thank the referee for the comment. The sentence referred to the results derived from analysis of cancer genomic data. We have now clarified this in the revised manuscript.

Author's action:

Updated sentence:

“The results derived from analysis of cancer genomic data probably apply to germ cells because they agree nicely with the observations from our mutational signature analysis with de novo mutations.”

2.8 Referee's comment:

In lines 515-517, it is stated that “(ii) recent studies suggested that replicative errors account for majority of mutations arising in both somatic and germ cells (Tomasetti and Vogelstein 2015; Tomasetti et al. 2017)”. However, there is still debate regarding the contribution of replication-error-induced mutations to germline mutations (see Gao et al 2019 for an counter-example). Even in the cancer field, most people believe that unrepaired DNA damage, in addition to replication errors, plays an important role in mutagenesis. In fact, the author's own mutational signature analysis suggests that Signature 1, presumably due to cytosine deamination, contributes a large fraction (>25%) of de novo mutations (Fig 3b). Therefore, the sentence mentioned above is incorrect and need to be revised. More generally, there is currently no consensus whether somatic and germline mutations are caused by similar mutational processes (and whether the same process has similar relative contributions).

Author's response:

We thank the referee for the comment. As it is still controversial in the field of mutagenesis, we have removed this sentence from the text.

Author's action:

Updated sentence:

“The results derived from analysis of cancer genomic data probably apply to germ cells because they agree nicely with the observations from our mutational signature analysis with de novo mutations.”

REVIEWERS' COMMENTS:

Reviewer #2 (Remarks to the Author):

The authors have addressed all points I raised in the previous round of review. I have no further major comments.

One minor question I have is about Fig 3: In Fig 3A, there is clear periodicity in sequence content (i.e., GC content) accompanying position of strong nucleosomes, so how do the authors exclude the possibility that the periodicity in mutation rates and escape ratios is due to sequence context rather than nucleosome positioning (lines 345-347)? A similar question also apply to the DSB signal in 3C: could the periodicity in DSB formation be due to sequence context instead of nucleosome positioning? I don't think this point is critical, but hope the authors could consider revise the text if the possibility exists.

REVIEWERS' COMMENTS:

Reviewer #2 (Remarks to the Author):

The authors have addressed all points I raised in the previous round of review. I have no further major comments.

One minor question I have is about Fig 3: In Fig 3A, there is clear periodicity in sequence content (i.e., GC content) accompanying position of strong nucleosomes, so how do the authors exclude the possibility that the periodicity in mutation rates and escape ratios is due to sequence context rather than nucleosome positioning (lines 345-347)? A similar question also apply to the DSB signal in 3C: could the periodicity in DSB formation be due to sequence context instead of nucleosome positioning? I don't think this point is critical, but hope the authors could consider revise the text if the possibility exists.

Author's response:

We thank the referee for the comments. We have revised the text to clarify our points.

We agree that the sequence context to some extent can explain the periodicity in mutation rates and escape ratios, but we think nucleosome positioning can contribute to the periodicity independent of sequence context. Notably, when looking at MMR-related mutation rates and escape ratios for A/T and C/G sites (Fig.4a), the profiles of A+T and G+C content show opposite trends, but the curves for mutation rates and escape ratios show similar periodical trends in both A/T and C/G sites (e.g. nucleosome dyads having lower values than linkers). In addition, the periodicity in profiles of A+T and G+C content almost disappears outside the central three nucleosomes, but for mutation rates and escape ratios the periodicity is still visible outside the central three nucleosomes. In the previous submission, we didn't exclude the possibility that sequence context contributes to the periodicity in mutation rates and escape ratios, as we wrote "suggestive of associations with nucleosome positioning rather than sequence alone". To make this point clearer, we revised the related sentences (see Author's action below).

Regarding the periodicity in DSB profiles, we also agree that it could be explained by sequence context. It may also be partly due to the library preparation protocol (e.g. reads may be under-sampled in linkers). Therefore, in the original manuscript we didn't mention the periodicity in the DSB section. We focused on comparing the values between strong nucleosomes and surrounding regions as well as the randomly selected regions. It should also be noted that without the END-seq data derived from naked DNA, it is difficult to assess the degree of the contribution of strong nucleosomes independent of the sequence context. We have clarified this point in the revised text by adding one sentence to the main text (see below).

Author's action:

We revised the following sentences:

"A/T sites have higher escape ratios than C/G sites around strong nucleosomes. Despite different nucleotide frequencies, both C/G and A/T sites exhibit similarly elevated escape ratio profiles (dyads having lower values than linkers; Fig. 4a),

suggesting that strong nucleosomes can contribute to the elevated escape ratios independent of sequence context. Moreover, the apparent ~200-bp periodicity in escape ratio and mutation density profiles are suggestive of associations with nucleosome positioning other than sequence context alone (Fig. 4a).”

We added the following sentence to the paragraph related to DSB:

“We also note that, because of the lack of END-seq data derived from naked DNA, it is difficult to assess the contribution of strong nucleosomes to the elevated DSB frequency independent of the sequence context.”